# Effect of PCSK9 Inhibitors on Regulators of Lipoprotein Homeostasis, Inflammation and Coagulation

**DOI:** 10.3390/biomedicines13020294

**Published:** 2025-01-24

**Authors:** Patricija Lunar, Hana Meglič, Mateja Vehar, Sabina Ugovšek, Andreja Rehberger Likozar, Miran Šebeštjen, Janja Zupan

**Affiliations:** 1Faculty of Medicine, University of Ljubljana, Vrazov trg 2, 1000 Ljubljana, Slovenia; patricija.lunar@gmail.com (P.L.); hana.meglic77@gmail.com (H.M.); ugovsek.sabina@gmail.com (S.U.); miran.sebestjen@guest.arnes.si (M.Š.); 2Faculty of Pharmacy, University of Ljubljana, Aškerčeva cesta 7, 1000 Ljubljana, Slovenia; mateja.vehar98@gmail.com; 3Department of Cardiology, University Medical Centre Ljubljana, Zaloška cesta 7, 1000 Ljubljana, Slovenia; 4Department of Vascular Diseases, University Medical Centre Ljubljana, Zaloška cesta 7, 1000 Ljubljana, Slovenia; andreja.rehbergerlikozar@kclj.si

**Keywords:** PCSK9 inhibitors, lipid regulators, inflammation, coagulation, coronary artery disease

## Abstract

Background: PCSK9 inhibitors (PCSK9i) represent a newer form of atherosclerosis treatment. Inflammation and haemostasis are key processes in the development of atherosclerosis. In this study, we investigated the influence of therapy with PCSK9i in patients with coronary artery disease (CAD) on regulators for lipoprotein homeostasis, inflammation and coagulation. Methods: Using quantitative polymerase chain reaction (qPCR), we measured the expression of the genes involved in lipoprotein homeostasis, namely for sterol regulatory element-binding protein 1 (*SREBP*1), *SREBP*2, low-density lipoprotein receptor (*LDLR*), hepatic lipase type C (*LIPC*), LDLR-related protein 8 (*LRP8*), and the genes associated with inflammation and coagulation, such as cluster of differentiation (CD) 36 (*CD*36), *CD*63, and *CD*14 in 96 patients with CAD and 25 healthy subjects. Results: Significant differences in the expression of the investigated genes between patients and healthy controls were found. Treatment with PCSK9i also resulted in significant changes in the expression of all studied genes. Conclusions: We established that PCSK9i may have a significant effect on the gene expression of lipid regulators, inflammatory markers, and coagulation parameters, independent of their lipolytic effect.

## 1. Introduction

Proprotein convertase subtilisin/kexin type 9 (PCSK9) inhibitors (PCSK9i) represent a new milestone in the treatment of the atherosclerotic process, particularly for the prevention of acute cardiovascular events [1]. The basic mechanism of their action is to reduce the concentration of low-density lipoprotein (LDL) cholesterol (LDL-C) by inhibiting the action of PCSK9 on the LDL-C receptor (LDLR) [2]. However, there is increasing research supporting their influence on platelet aggregation and inflammation, which presents important factors in the development and complications of the atherosclerotic process [3]. Rare patients with familial hypercholesterolemia (FH) that possess mutations in both the *PCSK*9 and *LDLR* gene have a significantly higher risk of future coronary heart disease (CHD) compared to FH patients bearing mutation in only one of the two genes [4].

Clearly, the influence of PCSK9i on other factors involved in the regulation of LDL-C concentration needs to be considered. Members of the sterol regulatory element-binding protein (SREBP) group play a central role in the regulation of genes important for lipid biosynthesis and uptake. In general, SREBP1 activates fatty acid and triglyceride synthesis, while SREBP2 initiates cholesterol synthesis [5]. The membrane-bound transcription factor SREBP2 is responsible for cholesterol homeostasis in cells and simultaneous gene expression of PCSK9 [6].

Lipase C hepatic type (LIPC) is involved in the metabolism of triglycerides and high-density lipoprotein cholesterol (HDL-C). At the same time, its increased activity leads to the formation of smaller and denser LDL-C particles, which have a greater ability to penetrate through the endothelial barrier and are significantly more atherogenic than larger and less dense LDL-C particles [7].

Apolipoprotein E receptor-2 (apoER2), the protein encoded by the *LRP*8 (LDL receptor-related protein-8) gene, is present in the cells involved in atherosclerosis pathogenesis such as platelets, endothelial cells [8], and monocytes/macrophages [9].

The binding of oxidized LDL (oxLDL) cholesterol to CD36 on the surface of platelets triggers a number of signalling pathways resulting in their activation, higher possibility of arterial thrombosis at the site of the atherosclerotic plaque rupture, and ultimately acute cardiovascular event [10].

CD63 is known as a platelet activation molecule, which plays a role in the processes of haemostasis and atherosclerosis. In platelets, CD63 is mainly found in α granules, which fuse upon activation with the plasmalemma and expose CD63 to the platelet surface [11].

Previous studies have demonstrated the importance of the interactions between CD14 and the proatherogenic factors in the development of atherosclerosis. Macrophages within atherosclerotic lesions were shown to express above-average levels of CD14. Triggering of the CD14-dependent signalling pathways increases the expression of the scavenger receptor-AI (SR-AI) and accelerates the formation of foam cells. CD14 signalling also affects the increased adhesion and migration of the inflammatory cells to the atherosclerotic lesions [12] and promotes cytokine expression and inflammation in macrophages. Increased levels of CD14 are associated with higher plasma cholesterol and a higher risk of development of CHD or other cardiovascular diseases [13].

The purpose of our study was to determine the effect of PCSK9i on the gene expression of the factors that participate, first, in the metabolism of atherogenic lipoproteins in addition to PCSK9, second, in the inflammation contributing to instability of the atherosclerotic plaques, and third, in the coagulation triggering the occurrence of thrombosis in the case of the atherosclerotic plaque rupture.

## 2. Materials and Methods

### 2.1. Patients and Controls

We included 96 high-risk patients after an acute coronary syndrome (ACS) who had, despite the maximal tolerated dose of statins, insufficiently regulated LDL-C levels. In addition, they had greatly increased levels of lipoprotein (a) (Lp(a)), which is an independent lipid risk factor regardless of the LDL-C levels and additional 25 healthy control subjects matched for sex and age. The inclusion of the patients and the study design is shown in Figure 1.

Inclusion and exclusion criteria are shown in Figure 2. All of the patients had been prescribed beta blockers and antiplatelet drugs and were receiving angiotensin-converting enzyme inhibitors/angiotensin II receptor blockers and statins at the highest tolerated doses, along with ezetimibe where needed. Their therapies had not been changed for at least 8 weeks before entering the study. Control subjects had no history of cardiovascular disease (CVD), no hypercholesterolemia, and Lp(a) levels less than 300 mg/L. Patients were randomly assigned to three groups: the first group (*n* = 34) received alirocumab 150 mg s.c. every 14 days, the second group (*n* = 32) received evolocumab 140 mg s.c. every 14 days, and the third group (*n* = 30) received placebo s.c. every 14 days.

### 2.2. Clinical Examination

Systolic and diastolic blood pressures were measured in the sitting position after a minimum of 10 min rest, with the mean of three measurements recorded. Anthropometric parameters were recorded, and body mass index was calculated.

### 2.3. Laboratory Analyses

The blood for laboratory analyses was taken in the morning after 12 h of fasting. Samples were collected from the antecubital vein into vacuum-sealed 5 mL tubes containing clot activator (Cacutube, LT Burnik, Komenda, Slovenia). Serum was obtained by 15 min centrifugation at 2000× *g*. Total cholesterol, triglycerides, high-density lipoprotein cholesterol, and apolipoproteins A1 and B were determined in the fresh serum by standard colorimetric or immunologic assays on an automated biochemistry analyser (Fusion 5.1; Ortho-Clinical Diagnostics, Raritan, NJ, USA). The Friedewald formula [14] was used to calculate LDL-C. The same biochemistry analyser was used to determine Lp(a) with the Denka reagent (Randox, London, UK), which contains apo(a) isoform-insensitive antibodies, and therefore showed minimal apo(a) size-related bias.

### 2.4. Gene Expression Measurement

The genes *SREBP*1, *SREBP*2, *LDLR*, *LIPC*, *LRP*8, *CD*36, *CD*63, and *CD*14 were measured in patients before and after PCSK9i or placebo treatment along with matched heathy controls. Total RNA was isolated from all samples collected in Tempus™ Blood RNA Tubes using a Tempus™ Spin RNA Isolation Kit (Thermo Fisher Scientific, Waltham, MA, USA) and evaluated as described previously [15]. Complementary DNA (cDNA) was synthesized using a High-Capacity cDNA Archive kit (Thermo Fisher Scientific, Waltham, MA, USA) according to the manufacturer instructions. Gene expression was measured according to MIQE guidelines [16]. Quantitative polymerase chain reaction (qPCR) was performed using 5× HOT FIREPol EvaGreen qPCR Supermix (Solis BioDyne, Tartu, Estonia) and LightCycler 480 II instrument (Roche, Basel, Switzerland). The sequences of the primers for the genes *CD*14, *LRP*8, *CD*36, *CD*63, and *SREBP*2 were obtained from online sources [17,18,19,20,21] and validated on our samples. The primers for genes *SREBP*1, *LDLR*, and *LIPC* (Appendix A) were self-designed and validated. The sequences for reference genes, glyceraldehyde-3-phosphate dehydrogenase (*GAPDH*), and ribosomal protein L13a (*RPL*13*A*) were taken from our previous studies. All of the data were normalized to the geometric mean of *GAPDH* and *RPL*13*A*.

### 2.5. Statistical Analysis

Statistical analysis was performed using IBM SPSS Statistics, version 27.0 (IBM Corporation, New York, NY, USA). First, the normality of the distribution of variables was tested using the Kolmogorov–Smirnov test. The values of variables that were normally distributed were presented with arithmetic mean and standard deviation. The median and the range between the lower and upper quartiles were used to display variables that were not normally distributed. For independent samples with normally distributed variables, Student’s *t*-test was used to test differences between the groups. The Mann–Whitney U test was used to test the differences between the groups with non-normally distributed variables. Differences between parameters at baseline and after 6 months of treatment were calculated using the Wilcoxon signed-rank test. Pearson and Spearman correlation analyses were used to determine the correlations between normally and non-normally distributed variables, respectively. The significance level was set to *p* < 0.05. For the gene expression analysis, the Bonferroni correction was considered and the significance level was set at * *p* < 0.01 and ** *p* < 0.001. GPower was used to perform the power of the study calculations [22]. The required sample size determined by using the a priori analysis (0.80 power of the study, 0.15 effect size, and 0.05 α error) was 22 subjects per group.

## 3. Results

### 3.1. Subjects’ Characteristics

Clinical and laboratory parameters of the patients and controls are shown in Table 1. Amongst clinical parameters, only body mass index (BMI) showed statistical significance between both groups, namely, healthy subjects had significantly lower BMI (*p* = 0.045). On the other hand, healthy subjects possessed significantly higher total cholesterol (*p* = 0.035), LDL-C (*p* = 0.023), and HDL-C (*p* = 0.034) than patients. We also detected significantly lower Lp(a) concentrations in healthy subjects compared to patients (*p* < 0.001).

Since there was no difference in any parameters in patients treated with PCSK9i or placebo before treatment, we decided to combine the groups for higher statistical power when comparing patients before treatment with healthy subjects.

### 3.2. The Results of Expression of the Tested Genes

As shown in Table 2, the expression of *SREBP*2, *LDLR*, *LIPC*, *LRP*8, *CD*36, *CD*63, and *CD*14 statistically significantly differed between patients and healthy subjects (** *p* < 0.001), except for *SREBP*1 (*p* = 0.976). *LDLR*, *LIPC*, and *LRP*8 expressions were significantly higher, while expressions of *SREBP*2, *CD*36, *CD*63, and *CD*14 were significantly lower in the patient group.

All data are normalized to the geometric mean of glyceraldehyde-3-phosphate dehydrogenase (GAPDH) and ribosomal protein L13a (RPL13A). Differences between the patients and controls were observed as indicated (** *p* < 0.001).

Next, we investigated the expression of the tested genes in patients before and after 6 months of treatment with PCSK9i alirocumab and evolocumab. We demonstrated that changes in the expression of all genes except *SREBP*1 and *LDLR* were statistically significant after 6 months of treatment. Gene expression of *LIPC* and *LRP*8 was significantly lower, whereas the expression of *SREBP*2, *CD*36, *CD*63, and *CD*14 was significantly higher (Figure 3 and Figure 4).

Furthermore, we analysed the expression of the investigated genes in the patient group receiving placebo. We found that the expression of *CD*36 had significantly increased (* *p* = 0.001).

Additionally, we compared the gene response after therapy with PCSK9i as opposed to placebo. Significantly different changes in the expression of *CD*36 (* *p* = 0.002) and *CD*63 (* *p* = 0.002) were observed, namely the expression of both genes was significantly lower in the group of patients receiving placebo (Figure 5 and Figure 6).

Additionally, we analysed changes in lipid profile after PCSK9i treatment, since alirocumab and evolocumab are medications for treatment of dyslipidaemias. In the placebo group, a minimal increase in total cholesterol concentrations (3%) was detected, while in PCSK9i-treated patients, a significant decrease (35%, ** *p* < 0.001) was observed. In the placebo and PCSK9i groups, the changes were as follows: 4% and −64% (** *p* < 0.001) for LDL-C, 7% and 8% (*p* = 0.725) for HDL cholesterol, 11% and −11% (* *p* = 0.005) for triglycerides, and 2% and −21% (** *p* < 0.001) for Lp(a).

Having identified the changes in lipid profile in patients treated with PCSK9i, we further aimed to investigate whether changes in lipid profile correlate with changes in the expression of the tested genes (Table 3). We found a borderline statistically significant negative correlation between change in HDL-C and change in *CD*63 (* *p* = 0.019; ρ = −0.294).

## 4. Discussion

To the best of our knowledge, this is the first study to investigate the effect of PCSK9i on factors participating in the metabolism of atherogenic lipoproteins, inflammation, and coagulation in high-risk CHD patients. The gene expressions of all investigated regulators of the lipid metabolism except for *SREBP*1 were significantly different in healthy controls compared to patients. The same is true for the gene expression of the regulators of inflammation and coagulation. At the same time, our results showed that treatment with PCSK9i changed the expression of all genes except *SREBP*1 and *LDLR*.

At first glance, it is surprising that healthy controls possessed increased values of total and LDL-C compared to patients; however, we have to bear in mind that all our patients were treated with the highest tolerated dose of statin and ezetimibe if needed. We examined the expression of several genes encoding the proteins involved in the LDL-C metabolism. SREBP-1 is responsible for both fatty acid and cholesterol synthesis, whereas SREBP-2 exclusively regulates cholesterol synthesis [23]. Drug-induced reduction in intracellular cholesterol levels triggers the activation of SREBPs, including SREBP-1 and SREBP-2, to restore the cholesterol balance [24]. Unfortunately, data for humans are not available. However, it was shown that high doses of statins in mice without hypercholesterolemia increase not only the production of 3-hydroxy-3-methylglutaryl coenzyme A (HMG-CoA) reductase, but also the gene expression for SREBP-2 [25]. This is in contrast to our results, as we found no differences in *SREBP*1 expression between healthy controls and patients, all of whom were treated with the highest tolerated doses of statins. However, *SREBP*2 expression was even higher in healthy controls than in patients. Yet, we have to be aware of two facts: first, the patients had significantly lower cholesterol levels compared to the healthy controls, which could have reversed the relationship in the case of *SREBP*1 and *SREBP*2, and second, all patients in our study had significantly increased Lp(a) levels before starting treatment with PCSK9i. It is well recognized that Lp(a) includes large amounts of apo(B), which might also have influenced these results.

In healthy individuals, a three-week low-carbohydrate/high-fat (LCHF) diet significantly increased both LDL-C and apo(B), as well as *SREBP1* gene expression. However, it is unclear whether the increase in *SREBP*1 gene expression and apo(B)-containing lipoproteins is causally related [26]. We have no data on the association between Lp(a) concentration and the gene expression for *SREBP*1 and 2.

In our study, treatment with PCSK9i did not alter *SREBP*1 expression, while there was a significant increase in *SREBP*2 expression. There are no clinical studies investigating the effect of PCSK9i on these proteins. However, the IB20 antibody against PCSK9 in mice reduced the gene expression of SREBP in hepatocytes consistent with a reduction in LDL-C levels. Consistent with this observation in mice, in statin-responsive human primary hepatocytes, IB20 lowers the transcription of several SREBP-regulated genes involved in cholesterol and fatty-acid synthesis [27]. These results are expected, as we assume that therapy with PCSK9i decreases intracellular cholesterol levels to the extent of increasing *SREBP2* transcription. Hence, our results are comparable to previous studies in mice.

Statins decrease plasma LDL-C by inhibiting HMG-CoA reductase, the rate-limiting enzyme in cholesterol synthesis. This inhibition induces intracellular cholesterol depletion, which results in upregulation of the hepatic LDLR [28]. Therefore, it is not unexpected that patients in our study had significantly higher *LDLR* expression compared to the control group. Interestingly, treatment with PCSK9i had no effect on *LDLR* expression, a result that is somewhat surprising, given that treatment with PCSK9i was presumed to lead to a significant reduction in LDL-C levels.

The *LIPC* gene encodes the enzyme hepatic lipase, which functions to hydrolyse triglycerides and phospholipids present in plasma lipoproteins. In addition, it acts as a ligand to cellular receptors and proteoglycans on the surface of hepatocytes to facilitate the uptake of lipoproteins into the liver. The results of previous studies in humans and animals demonstrate both the proatherogenic and antiatherogenic roles of hepatic lipase. Its proatherogenic effects are the result of an inverse relationship between increased hepatic lipase activity and plasma concentrations of antiatherogenic HDL-C, and a positive correlation with plasma concentrations of proatherogenic LDL-C. The lipolytic function of hepatic lipase, which leads to lower levels of proatherogenic apolipoprotein B-containing lipoproteins in plasma, reduces the risk of developing atherosclerosis [29]. The influence of hepatic lipase activity on the risk of CHD is not clear and mainly depends on the concentration and type of lipoprotein(s) that predominate in a particular lipid metabolic disorder [30]. In general, serum hepatic lipase concentrations are very low, as most hepatic lipase is bound to proteoglycans on the surface of hepatocytes, and high serum concentrations are therefore considered a risk factor for the development of CHD [31]. Several studies have demonstrated significantly higher serum hepatic lipase levels in patients with CHD compared to healthy subjects, suggesting that serum hepatic lipase levels could be used as a potential marker of coronary artery disease progression [31,32]. Given the above, it is not surprising that the patients in our study had significantly higher *LIPC* expression compared to the control group. Our results also showed that after six months of treatment with PCSK9i, lipid levels decrease to such an extent that *LIPC* expression also decreases. However, the results of numerous studies regarding the role of hepatic lipase in lipid homeostasis are contradictory, suggesting its proatherogenic and antiatherogenic role. Based on the results of our study, we merely suggest that a positive correlation between lipid concentration and *LIPC* gene expression exists.

Similar to our study, Shen et al. found that *LRP*8 expression is significantly increased in patients with premature coronary artery disease compared to healthy individuals [33]. Unfortunately, this study lacks information on whether the patients were previously treated with statins and, if so, at what dose. To the best of our knowledge, there are no other studies evaluating the impact of statin treatment on *LRP8* expression. We were also not able to find any data on the association between LRP8 and PCSK9 in the literature. Considering that Shen et al. reported increased LRP8 levels in patients with premature coronary artery disease [33], we could assume that the downregulation of *LRP*8 expression observed after treatment with PCSK9i in our study may have a beneficial effect on inhibiting the atherosclerotic process.

CD36 expression at the protein level has been shown to be significantly increased in patients with carotid atherosclerosis, particularly in the advanced stages of the disease [34]. All of our patients had an advanced form of atherosclerosis, as they had all suffered from ACS. However, our results showed lower *CD*36 expression in patients compared to healthy subjects. This discrepancy suggests that the gene and protein levels may differ. Increased expression of surface marker *CD*36 is thought to activate macrophages, leading to the formation of unstable atherosclerotic plaques that are more prone to rupture, thereby resulting in subsequent thrombosis [35]. Additionally, increased CD36 levels contribute to accelerated oxidation of LDL-C, which is considered to be even more atherogenic than LDL-C itself [36]. Treatment with statins has been shown to decrease the CD36 expression on platelet surfaces, thus inhibiting the progression of atherosclerosis. This reduction occurs independently of LDL-C lowering, suggesting the so-called pleiotropic effects of statins [37]. In our study, treatment with both PCSK9i and placebo significantly increased *CD*36 expression. Given the established association between increased *CD36* expression on platelets and the progression of the atherosclerotic process [34], one might speculate that treatment with PCSK9i does not appear to beneficially affect the process of atherosclerosis. However, it is important to note that, in our study, *CD*36 expression was measured at the gene level, which might not accurately reflect the levels of the CD36 protein on the platelet surfaces. On the other hand, *CD*36 expression in our healthy subjects was higher than in patients receiving either PCSK9i or placebo. Of note, *CD36* expression was comparable between healthy controls and patients after PCSK9i treatment, but this was not the case for the patients receiving placebo. One potential mechanism by which PCSK9i may influence CD36 production could involve inhibition of its degradation in hepatocytes and adipocytes. While PCSK9 has been shown to promote CD36 degradation in mouse hepatocytes [38], data on its effects in humans are lacking.

Moreover, the results of our study showed significantly higher *CD*63 expression in healthy controls compared to patients with CHD. In contrast, Murakami et al. using flow cytometry found no differences in *CD*63 expression on platelets between healthy controls and patients with CHD [39]. At the same time, they found no correlation between *CD63* expression and the extent of coronary disease. However, it is important to note that the subjects in their study were not treated with statins prior to the study. Cha et al. demonstrated significantly higher CD63 expression on platelets in patients with hyperlipidaemia after ischemic stroke compared to healthy controls. Treatment with simvastatin in these patients resulted in significant reduction in CD63 expression on platelets [40]. Their findings are similar to ours, as all the patients in our study had been previously treated with statins and had significantly lower *CD*63 expression than healthy controls. There is currently no data on the association between CD63 and PCSK9 levels, and even less on the potential effects of treatment with PCSK9i on either the protein levels or the expression of the *CD*63 gene. In line with the expression of the *CD*36 gene, our study showed that treatment with PCSK9i led to a significant increase in the expression of the *CD*63 gene. Similar to *CD*36, *CD*63 gene expression in the control group was comparable to that of patients treated with PCSK9i, but not to those who received the placebo.

In patients with CHD who had not previously been treated with statins, CD14^+^ monocyte expression was reduced compared to healthy controls [41]. Although all of our CHD patients had received statin treatment prior to study, they also had lower *CD*14 expression. In our study, treatment with PCSK9i led to an increase in *CD*14 expression, whereas no statistically significant changes were observed in the placebo group. Given that CD14 is expressed on monocytes, which typically have an opposing effect, it is difficult to suspect a potential antiatherosclerotic effect of PCSK9i. It is important to consider that the role of the monocytes in the atherosclerosis is strongly influenced by both surface markers CD14 and CD16, as well as their ratio [42]. However, since CD14 levels are generally reduced in patients with advanced atherosclerosis [41], the observed increase in *CD*14 expression might have a beneficial effect on the atherosclerotic process. In patients with stable coronary artery disease, a statistically significant correlation was found between monocyte concentration and Lp(a)-PCSK9 complex concentration, mainly due to monocytes expressing the highest CD14 levels on their surface and exhibiting the most pronounced pro-inflammatory effects [43]. This association might have been even more significant in our patients, who had highly elevated Lp(a) levels prior to treatment. Despite a significant decrease in Lp(a) following treatment with PCSK9i, their Lp(a) levels remained considerably higher than the desired target levels according to current treatment guidelines [44]. Furthermore, oxidized phospholipids, which are a component of Lp(a) and have a strong pro-inflammatory effect, are significantly associated with the concentration of pro-inflammatory monocytes expressing high levels of CD14 on their surface [45].

At a first glance, it is surprising that we did not find differences in the expression of genes encoding lipoprotein homeostasis regulators between patients receiving PCSK9i and placebo. Both groups of patients already had relatively low LDL-C levels before therapy, which were further significantly reduced in the group of patients treated with PCSK9i. One reason for the lack of differences in the expression of genes encoding lipoprotein homeostasis regulators could be that there is a threshold in the intracellular LDL-C concentration below which no influence on the expression of genes encoding lipoprotein homeostasis regulators could be detected. The significant increased expression of *CD36* after placebo treatment is also surprising. However, the gene expression and protein translation are complex processes influenced by different factors that do not depend solely on lipoprotein concentration. As for the rest of the significant results demonstrated in this study, their further verification on protein level is required to underpin these findings.

In our study, no statistically significant associations were found between the change in gene expression and lipids. We observed only borderline significant association of the change in *CD63* gene expression with HDL-C. All other changes could be attributed to pleotropic effects, i.e., effects independent on the effect on lipid parameters. In addition to lowering LDL-C, PCSK9i also lowered Lp(a) by 20–35%, but the mechanism of the latter is not fully elucidated [46]. The peculiarity of our study is, as previously mentioned, that the patients experienced an expected decrease in Lp(a) concentration by approximately 20%, but at the end of the study, their Lp(a) concentrations were still strongly in the atherogenic range. This finding is not surprising given that only patients with extremely elevated Lp(a) values were included in our study. The atherogenic values of Lp(a) could imply a proatherogenic effect due to the similarity of Lp(a) with LDL-C, a prothrombotic effect due to similarity between apo(a) and plasminogen, and a pro-inflammatory effect due to OxPLs in Lp(a) [22]. On the other hand, the LDL-C levels in our patients were very low, as all patients reached the concentrations below 1.4 mmol/L at the end of the study. Given that drugs that specifically reduce Lp(a) concentration by up to 90% are currently in the clinical trial phase (NCT04023552), it would be interesting to see whether the addition of these drugs to our patients would have an additional effect on the expression of the studied genes as well as on the reduction in proatherogenic, prothrombotic, and pro-inflammatory effects of Lp(a) leading to a possible reduction in acute cardiovascular events.

There are two shortcomings of this study. The first one is the relatively small number of subjects, which is a consequence of the very strict inclusion criteria, which can, conversely, be an advantage due to the very homogeneous population. The second limitation is that all the measurements were performed at the level of gene expression, which does not necessarily correlate with the levels of proteins that are a result of the expression of these genes. The process of translation, i.e., gene expression until the formation of a functional protein, is influenced by a large number of factors that are mostly unknown and therefore could not be taken into account in our analyses. Hence, the additional research with larger cohorts and protein level analyses is needed to further solidify these findings and their clinical relevance. Moreover, the mechanisms beyond PCSK9i influence on lipid regulators, inflammatory markers, and coagulation parameters identified here need to be identified to pave the way for new possible treatment strategies.

## 5. Conclusions

Inflammation and haemostasis, as crucial processes in the development of atherosclerosis, are targets of the novel treatment with PCSK9i. Our study observed a significant difference in the gene expression of the factors playing a role in lipoprotein homeostasis, inflammation, and coagulation between healthy subjects and patients. Moreover, we demonstrated that treatment with PCSK9i influences the gene expression of most of these factors. However, gene expression of *CD*36 also significantly changed after placebo, but was significantly lower compared to therapy with PCSK9i. Given that the primary effect of PCSK9i is to reduce LDL-C and, to a lesser extent, Lp(a), we can argue that the effects on the expression of the studied genes are independent of the lipolytic effect of PCSK9i. Of course, to confirm that these results are also clinically significant, a study with a much larger number of subjects would be needed, where, in addition to gene expression, the result of their expression at the protein level would also be measured in relation to the effects on cardiovascular events. To evaluate the influence of the significant results shown here on the plaque stability and the reduction in cardiovascular events, a longer prospective clinical study is needed.

## Figures and Tables

**Figure 1 biomedicines-13-00294-f001:**
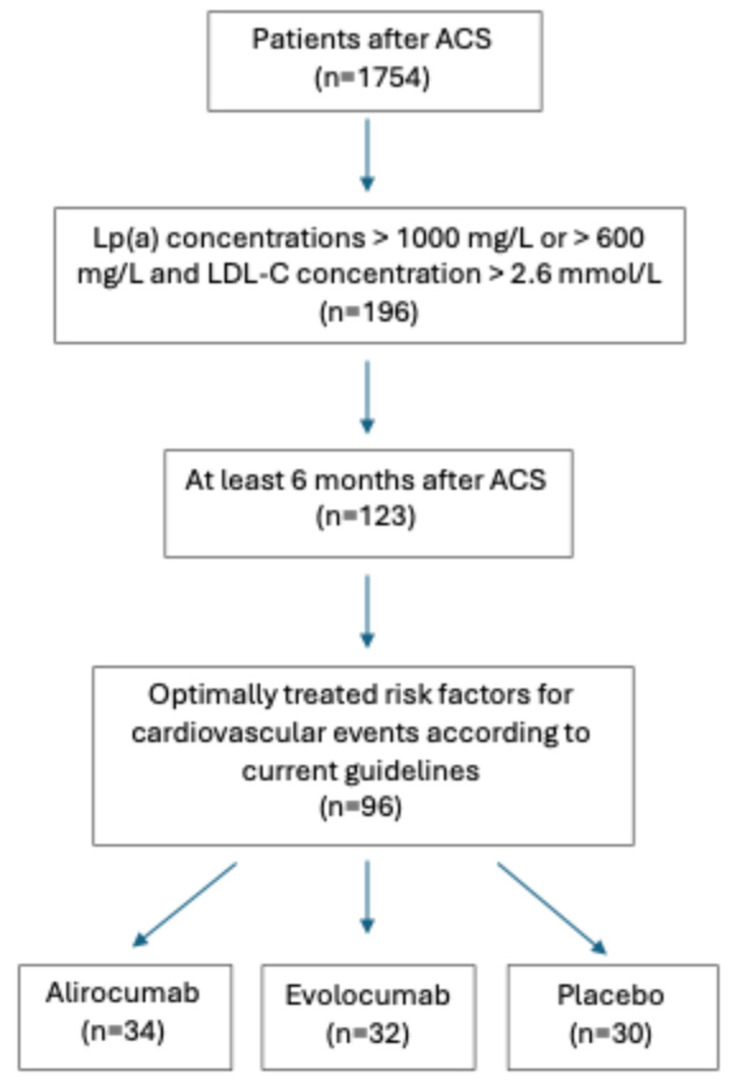
The inclusion of the patients and the study design. ACS—acute coronary syndrome. LDL-C—low-density lipoprotein cholesterol. Lp(a)—lipoprotein(a).

**Figure 2 biomedicines-13-00294-f002:**
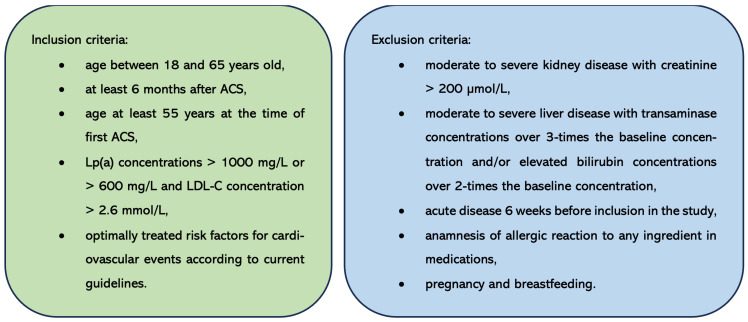
Inclusion and exclusion criteria for the patient cohort. ACS—acute coronary syndrome. LDL-C—low-density lipoprotein cholesterol. Lp(a)—lipoprotein(a).

**Figure 3 biomedicines-13-00294-f003:**
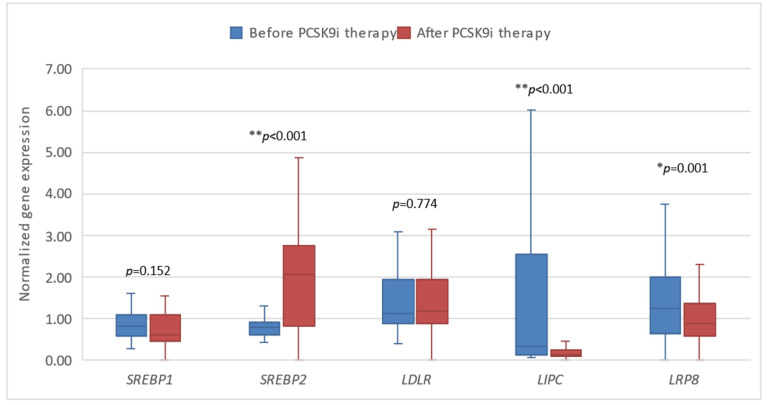
The expression of genes encoding lipoprotein homeostasis regulators in patients before and after therapy with PCSK9i. *SREBP*—gene encoding sterol regulatory element-binding protein. *LDLR*—gene encoding LDL receptor. *LIPC*—gene encoding hepatic lipase type C. *LRP*8—gene encoding LDLR-related protein 8. Differences before and after PCSK9i therapy were observed as indicated (** *p* < 0.001, * *p* < 0.01).

**Figure 4 biomedicines-13-00294-f004:**
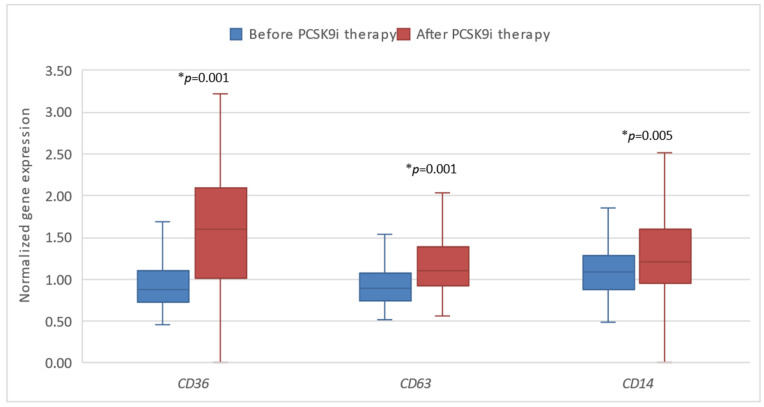
The expression of genes encoding inflammatory factors in patients before and after therapy with PCSK9i. *CD*36—gene encoding cluster of differentiation 36. *CD*63—gene encoding cluster of differentiation 63*. CD*14—gene encoding cluster of differentiation 14. Differences before and after PCSK9i therapy were observed as indicated (* *p* < 0.01).

**Figure 5 biomedicines-13-00294-f005:**
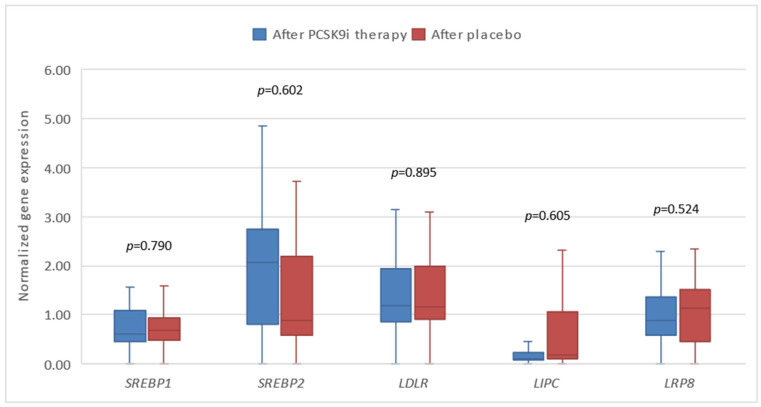
The expression of genes encoding lipoprotein homeostasis regulators in patients after therapy with PCSK9i and after placebo. *SREBP*—gene encoding sterol regulatory element-binding protein. *LDLR*—gene encoding LDL receptor. *LIPC*—gene encoding hepatic lipase type C. *LRP*8—gene encoding LDLR-related protein 8.

**Figure 6 biomedicines-13-00294-f006:**
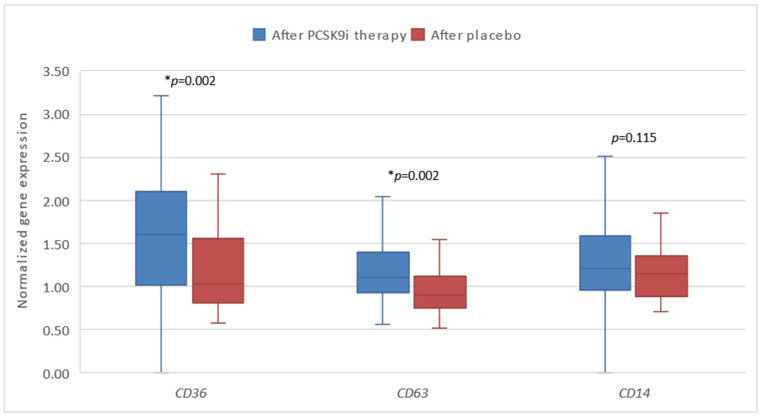
The expression of genes encoding inflammatory factors in patients after therapy with PCSK9i and after placebo. *CD*36—gene encoding cluster of differentiation 36. *CD*63—gene encoding cluster of differentiation 63. *CD*14—gene encoding cluster of differentiation 14. Differences before and after PCSK9i therapy were observed as indicated (* *p* < 0.01).

**Table 1 biomedicines-13-00294-t001:** Clinical and lipid parameters in patients and healthy controls.

	Patients (*n* = 96)	Controls (*n* = 25)	*p* Value
Gender (M/F)	91/5	23/2	0.641
Age (years)	50.46 ± 8.74	48.92 ± 7.14	0.076
BMI (kg/m^2^)	28.57 ± 3.79	25.40 ± 3.32	* 0.045
Total cholesterol [mmol/L]	4.25 ± 0.88	5.80 ± 0.67	* 0.034
LDL-C [mmol/L]	2.33 ± 0.77	3.56 ± 0.63	* 0.023
HDL-C [mmol/L]	1.17 ± 0.27	1.52 ± 0.42	* 0.034
TG [mmol/L]	1.47 (1.04–2.10)	1.31 (0.99–2.01)	0.096
Lp(a) [mg/L]	1431.00 (1203.00–1658.00)	11.00 (4.00–18.50)	** <0.001

BMI—body mass index. M/F—male/female. LDL-C—low-density lipoprotein cholesterol. HDL-C—high-density lipoprotein cholesterol. TG—triglycerides. Lp(a)—lipoprotein(a). Differences between the patients and controls were observed as indicated (** *p* < 0.001, * *p* < 0.05).

**Table 2 biomedicines-13-00294-t002:** The results of gene expression in patients and controls.

	Patients (*n* = 96)	Controls (*n* = 25)	*p* Value
*SREBP*1	0.82 (0.60–1.11)	0.80 (0.49–1.72)	0.976
*SREBP*2	0.77 (0.63–0.89)	2.33 (1.90–2.87)	** <0.001
*LDLR*	1.18 (0.89–2.24)	1.32 (1.04–2.33)	** <0.001
*LIPC*	0.35 (0.14–2.60)	0.08 (0.06–0.11)	** <0.001
*LRP*8	1.56 (0.94–2.73)	0.75 (0.52–1.20)	** <0.001
*CD*36	0.86 (0.72–1.02)	1.69 (1.34–2.14)	** <0.001
*CD*63	0.87 (0.76–1.02)	1.23 (1.03–1.48)	** <0.001
*CD*14	1.07 (0.85–1.25)	1.31 (1.13–1.74)	** <0.001

SREBP—gene encoding sterol regulatory element-binding protein. LDLR—gene encoding LDL receptor. LIPC—gene encoding hepatic lipase type C. LRP8—gene encoding LDLR-related protein 8. CD36—gene encoding cluster of differentiation 36. CD63—gene encoding cluster of differentiation 63. CD14—gene encoding cluster of differentiation 14 (** *p* < 0.001).

**Table 3 biomedicines-13-00294-t003:** The correlations between changes in lipid profile and the expression of the investigated genes.

	Δ*SREBP*1	Δ*SREBP*2	Δ*LDLR*	Δ*LIPC*	Δ*LRP*8	Δ*CD*36	Δ*CD*63	Δ*CD*14
ΔTC	ρ = 0.252 *p* = 0.045	ρ = −0.077 *p* = 0.544	ρ = −0.023 *p* = 0.859	ρ = 0.163 *p* = 0.198	ρ = −0.062 *p* = 0.641	ρ = 0.072 *p* = 0.576	ρ = 0.218 *p* = 0.086	ρ = −0.065 *p* = 0.610
ΔLDL-C	ρ = 0.082 *p* = 0.521	ρ = −0.031 *p* = 0.806	ρ = −0.136 *p* = 0.282	ρ = 0.091 *p* = 0.477	ρ = −0.015 *p* = 0.910	ρ = 0.144 *p* = 0.263	ρ = 0.230 *p* = 0.069	ρ = 0.050 *p* = 0.694
ΔHDL-C	ρ = −0.094 *p* = 0.461	ρ = −0.075 *p* = 0.554	ρ = −0.043 *p* = 0.734	ρ = −0.172 *p* = 0.173	ρ = −0.123 *p* = 0.354	ρ = −0.055 *p* = 0.672	ρ = −0.294 *p* = 0.019	ρ = −0.231 *p* = 0.066
ΔTG	ρ = 0.108 *p* = 0.397	ρ = −0.043 *p* = 0.735	ρ = 0.089 *p* = 0.483	ρ = −0.044 *p* = 0.728	ρ = −0.156 *p* = 0.237	ρ = −0.081 *p* = 0.531	ρ = 0.179 *p* = 0.160	ρ = −0.021 *p* = 0.871
ΔLp(a)	ρ = −0.013 *p* = 0.921	ρ = −0.096 *p* = 0.457	ρ = −0.002 *p* = 0.985	ρ = 0.098 *p* = 0.450	ρ = 0.050 *p* = 0.715	ρ = 0.105 *p* = 0.427	ρ = 0.187 *p* = 0.149	ρ = 0.053 *p* = 0.684

Values with the symbol **Δ** represent the differences between values at the beginning of the study and after 6 months of PCSK9i treatment. The Δ values were calculated using the following equation: ((value after 6 months-value before treatment)/value before treatment) × 100%. Values are expressed as correlation coefficient (ρ) and *p* value. TC—total cholesterol. LDL-C—low-density lipoprotein cholesterol. HDL-C—high-density lipoprotein cholesterol. TG—triglycerides. *SREBP*1*/*2 –sterol regulatory element-binding protein 1/2 gene. *LDLR*—LDL receptor gene. *LIPC*—hepatic lipase type C gene. *LRP*8—LDLR-related protein 8 gene. *CD36—*cluster of differentiation 36 gene. *CD*63—cluster of differentiation 63 gene. *CD*14—cluster of differentiation 14 gene.

## Data Availability

The original contributions presented in this study are included in the article/Appendix A. Further inquiries can be directed to the corresponding author.

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
