# Peer review of "Effect of PCSK9 Inhibitors on Regulators of Lipoprotein Homeostasis, Inflammation and Coagulation"

_biomedicines, 2025, doi:10.3390/biomedicines13020294_

Round 1
Reviewer 1 Report
Comments and Suggestions for Authors
I am grateful to the editor for the opportunity to review the manuscript by Patricija Lunar et al. "Effect of PCSK9 inhibitors on regulators of lipoprotein homeostasis, inflammation and coagulation". In this article, the authors studied for the first time the effect of PCSK9i on factors involved in the metabolism of atherogenic lipoproteins, inflammation and coagulation in patients with high-risk coronary artery disease. A distinctive feature of this study is that it examined the gene expression of all the studied regulators of lipid metabolism, as well as regulators of inflammation and coagulation. The data are very interesting, but further studies with a larger number of patients will be needed to confirm their clinical significance, as well as on the effect of the observed changes on cardiovascular events.
While reviewing, I had the following comments and questions:
1. The aim of the study in the Introduction section is not formulated quite correctly. I think that the first sentence in the last paragraph (lines 77-81) would be enough for this. The following sentences (lines 81-87) would be more appropriate to place in section 2. Materials and Methods.
2. In section 2.1. Patients and controls, the characteristics of the included patients are insufficient ("We included 96 patients ..." - line 90). Here, it is necessary to provide at least information about the nosological composition of the patients (for example, the presence of chronic coronary syndrome), as well as the information that the authors placed in the Introduction section.
3. In section 2.5. Statistical analysis, the authors indicate that "The differences between the three groups were calculated using one-way ANOVA or Kruskal-Wallis test for the non-normally distributed variables." (lines 148-149). However, the authors do not provide data comparing the three groups in the article; an explanation of the absence of this data is needed.
4. It is necessary to add a flow chart indicating the inclusion of patients and the study design.
5. In the text of the manuscript, in the Results section, data on the group of patients who received placebo are provided. However, it remains unclear how many patients were in the PCSK9i treatment group, how many in the placebo group, how much these groups differed in baseline parameters (laboratory and expression of the studied genes), how the patients were randomized into groups. 6. Interestingly, the groups did not differ in expression of genes encoding lipoprotein homeostasis regulators’ in patients after therapy with PCSK9i and after placebo (Figure 4). How can this finding of the authors be explained?
Reviewer 2 Report
Comments and Suggestions for Authors
Lunar et al. reported their work titled “Effect of PCSK9 inhibitors on regulators of lipoprotein homeostasis, inflammation and coagulation” and concluded that “we established that PCSK9i may have a significant effect on the gene expression of lipid regulators, inflammatory markers and coagulation parameters, independent of their lipolytic effect.” I have the following comments:
-
Study Design and Sample Size: The study design is well-structured, but the sample size is relatively small, especially for the control group (n=25). This may limit the generalizability of the findings. A larger cohort would strengthen the statistical power and validity of the results. Please report the current statistical power.
-
Gene Expression vs. Protein Levels: The study focuses on gene expression, but it does not measure protein levels or functional outcomes. Since gene expression does not always correlate with protein activity, it would be beneficial to include protein-level data to confirm the biological relevance of the observed changes (Optional request).
-
Placebo Group Comparison: The study reports significant changes in gene expression in the placebo group for SREBP2, LRP8, and CD36. This raises questions about the specificity of the PCSK9i effects. A more detailed discussion on potential placebo effects or other confounding factors is needed.
-
Clinical Relevance: While the study demonstrates changes in gene expression, the clinical significance of these changes is not fully explored. It would be valuable to discuss how these changes might translate into clinical outcomes, such as reduced cardiovascular events or plaque stability.
-
Lipoprotein(a) Levels: The study highlights the reduction in Lp(a) levels with PCSK9i treatment, but the levels remained in the atherogenic range. It would be interesting to discuss the implications of this finding and whether additional therapies targeting Lp(a) could further enhance the benefits of PCSK9i.
-
Mechanistic Insights: The study could benefit from a deeper mechanistic exploration of how PCSK9i influences the expression of the studied genes. For example, are these effects direct or mediated through other pathways?
-
Statistical Analysis: The statistical analysis is robust, but the use of multiple comparisons increases the risk of Type I errors. Consideration of adjustments for multiple testing (e.g., Bonferroni correction) would be prudent, in the case of normally distributed variables.
-
Discussion of Conflicting Results: Some findings, such as the increase in CD36 expression with PCSK9i treatment, seem counterintuitive given its proatherogenic role. The discussion should address these discrepancies and provide potential explanations.
Additional minor comments: Please add background, methods, results, and conclusion subtitles to your abstract. Please specify the number of patients and control in the abstract.
Overall, the study provides valuable insights into the pleiotropic effects of PCSK9i, but additional research with larger cohorts and protein-level analyses would further solidify the findings and their clinical relevance.
Reviewer 3 Report
Comments and Suggestions for Authors
The manuscript is focused on therapeutical effects of PCSK9 inhibitors on expression of genes involving in lipoprotein homeostasis, inflammatory markers and coagulation parameters in patients with high-risk coronary artery disease (CAD).
In the study, 96 CAD patients and 25 healthy control persons were included. Blood pressures and anthropometric parameters including body mass index were examined and calculated. Detail analysis of therapy in CAD patients were collected.
Laboratory blood analysis of lipid metabolism was performed. Total RNA was isolated for analysis of gene expression (CD14, LRP8, CD36, CD63, SREBP2, SREBP1, LDLR, LIPC, GAPDH, RPL13a). With exception of SREBP1, expressions of all other genes were significantly higher in CAD patients compared to control persons.
The approach was repeated after 6 months of therapy by PCSK9i. Significant changes in gene expression due to the therapy except SRBP1 and LDLR were found.
PCSK9 inhibitors were shown to have high potential in the treatment of atherosclerotic process, especially useful they are in prevention of acute cardiovascular events. The study brings many laboratory data about their effects on lipoprotein homeostasis, inflammation and hemostasis.
It is very useful clinically oriented study, informative for medicine doctors, clearly written.
Round 2
Reviewer 1 Report
Comments and Suggestions for Authors
The authors answered my questions and comments, made corrections to the text of the manuscript. I have no other comments.
Reviewer 2 Report
Comments and Suggestions for Authors
Thanks for addressing my prior comments and it is my pleasure to accept this work.